# Minimizing Long-Term Toxicities for Patients with Primary Mediastinal B-Cell Lymphoma Undergoing Modern Radiotherapy: Results from a Monocentric Biophysical Risk Evaluation

**DOI:** 10.3390/cancers16244265

**Published:** 2024-12-22

**Authors:** Andrea Baehr, Sebastian Schäfer, Maria Jäckel, Saskia Alexandra Becker, Susanne Ghandili, Maximilian Grohmann, Hans Theodor Eich, Michael Oertel

**Affiliations:** 1Department of Radiation Oncology, University Hospital of Hamburg-Eppendorf, 20246 Hamburg, Germany; 2Department of Radiation Oncology, University Hospital Leipzig, 04103 Leipzig, Germany; 3Radiologische Allianz Hamburg, 20354 Hamburg, Germany; 4II. Medical Department and Clinic, Department of Oncology, Hematology and Bone Marrow Transplants with the Section Pulmonology, University Medical Center Hamburg-Eppendorf, 20246 Hamburg, Germany; 5Department of Radiation Oncology, University Hospital of Münster, West German Cancer Center (WTZ), Network Partner Site, 48149 Münster, Germany

**Keywords:** primary mediastinal lymphoma, planning study, normal tissue complication probability (NTCP)

## Abstract

For young patients with the rare entity primary mediastinal B-cell lymphoma, a differentiated description of effects and side effects of the different therapeutic opportunities is of immense interest. We therefore conducted a planning study to compare two different radiation therapy planning variants for a cohort of these patients who were treated in a hospital cancer center. We estimated normal tissue complication probabilities for radiation side effects for the heart, the lungs and the esophagus.

## 1. Introduction

Primary mediastinal B-cell lymphoma (PMBCL) is a rare form of aggressive B-cell lymphoma with a predominant onset in young female patients, with a median age of 35 years [1]. Standard systemic therapy consists of polychemotherapy with cyclophosphamide, hydroxydaunorubicin, vincristine and prednisone in combination with rituximab (R-CHOP) or an intensified modification with added etoposide (DA-EPOCH-R) [1,2,3].

The role of radiotherapy (RT) for PMBCL is vividly discussed, as databank and retrospective analyses elaborated conflicting results regarding its potential benefit [4,5,6,7]. Notably, a recent FDG-PET-based analysis from British Columbia revealed no deterioration in time to progression or overall survival (OS) when RT was limited to patients with positive end-of-treatment PET-CT [7]. Some authors even favor the advantages of serial PET-CT imaging after chemotherapy instead of adjuvant RT [8].

Regarding prospective trials, a subgroup analysis of the UNFOLDER trial by the German High-Grade Non-Hodgkin’s Lymphoma Study Group for patients with intermediate-prognosis aggressive B-cell lymphoma demonstrated improvement in the primary endpoint event-free survival rate with the use of RT (3 year rate: 94% versus 78%), with no amelioration of progression-free survival or OS [9]. To further evaluate the role of RT, dose-intensified immunochemotherapy approaches have been developed to omit additional radiation treatment [10]. Current guidelines recommend RT for consolidation in case of PET-positive residuals after immunochemotherapy [2] or only after biopsy confirmation [3].

With most patients with PMBCL being young, minimizing potential (long-term) side effects is of cardinal interest. Although data on cardiopulmonary sequelae are limited, a priori anticipation of the individual risk profile is desirable to counsel a patient on the putative impact of RT. Previous works demonstrated the feasibility of normal tissue complication probability calculations for hematological malignancies concerning thoracic organs at risk [11,12]. However, these analyses often included variations in case numbers and disease involvement, which may have hampered generalization.

To provide further evidence, we analyzed a cohort of 25 patients with homogenous disease extents to estimate the cardiopulmonary and esophageal risk attributable to RT. In addition, alternative plans with different techniques were designed for comparison. We further discuss both the physical and (radio)oncological aspects of the RT planning process for this complex entity.

## 2. Methods

This planning study included CT data from 25 patients from a hospital cancer center who were treated between the years 2015 and 2024. Among all patients, 6 received radiation therapy, whereas 19 went to follow-up after completion of chemotherapy. The patient characteristics relevant for this planning study are shown in Table 1. For all patients, two RT plans were prospectively designed for a TrueBeam linear accelerator (Varian Medical Systems, Palo Alto, CA, USA): one with two parallel opposing fields (APPA) and another with volume-modulated arc therapy (VMAT) technique. All plans were derived from PET-CT-guided contouring based on a CT scan after completion of chemotherapy (R-CHOP, R-CHOEP, daEPOCH-R or R-CHOEP-Brut). The contouring was aligned with literature recommendations [13]. The gross tumor volume (GTV) included the post-chemotherapy lymphoma residuum based on CT and PET-CT imaging. The CTV was designed as an individual volume to address subclinical spread for each patient based on post-chemotherapy CT scans. The planning target volume (PTV) was defined with a margin of 0.5 to 1 cm around the CTV adapted to the anatomic structures. The contouring was carried out by a radiation oncologist and supervised by an experienced senior radiation oncologist. The prescribed dose was 40 Gy in 2 Gy fractions for each plan.

All plans were calculated to meet the following planning objectives for CTV and PTV: CTV V95% > 98%, PTV D50% = 100% ± 1%, PTV V95% > 90% (The aim was V95% > 95%, but due to an occasional relatively large overlap between the PTV and the lungs, achieving V95% > 95% was not always possible.) and PTV V107% < 2%. Vx% denotes the percentage of the volume covered by x% of the prescription dose, while Dy refers to the dose received by y% of the volume. The dose to organs at risk (OARs) was reduced as much as possible, following the as low as reasonably achievable (ALARA) principle, without compromising PTV coverage. The planning conformity was described using the Paddick conformity index, which describes how accurate the prescription isodose conforms to the target, with high indices indicating high plan conformity [14].

All plans were generated using Eclipse Treatment Planning System v. 16.1 (Varian Medical Systems, Palo Alto, CA, USA). The APPA plans used beam energies of 6 MV and 15 MV and were calculated with the analytical anisotropic algorithm, whereas the VMAT plans used a beam energy of 6 MV and were calculated with Acuros^®^ XB Advanced Dose Calculation (AcurosXB in the following). Depending on the size and location of the PTV, one or two full or partial arcs were utilized in the VMAT plans to minimize the dose to organs at risk.

### Normal Tissue Complication Probability (NTCP) Calculation

A variety of dose-response and normal tissue toxicity models exist, with the Lyman–Kutcher–Burman (LKB) model being one of the most widely used models for radiotoxicity modeling [15]. It stipulates a sigmoidal response of normal tissue to radiation and relies on three parameters: TD_50_, the dose corresponding to a normal tissue complication probability of 50%, the slope m of the dose–response curve at the TD_50_ point and a parameter n, which is related to the volume effect. As modern radiotherapeutic treatments allow for highly complex dose distributions, which often involve partial irradiation of OARs, the equivalent uniform dose (EUD) concept was used to make the different dose-volume histograms (DVHs) comparable. The EUD for a given organ can be calculated as follows:$$EUD={\left(\underset{i}{\sum }v_{i}\cdot {d_{i}}^{1/n}\right)}^{n}$$
where $v_{i}$ is the volume and $d_{i}$ is the dose at point i of a differential DVH.

The NTCP for a given OAR was then calculated using the following equation:$$NTCP=\frac{1}{\sqrt{2\pi }}\overset{t}{\underset{-\infty }{\int }}e^{-\frac{x^{2}}{2}}dx$$
with t being given by
$$t=\frac{EUD-TD_{50}}{m\cdot TD_{50}}$$

For computational efficiency, the Gaussian cumulative density distribution function was approximated using the following function:$$NTCP={\left(e^{\left(-358\cdot \frac{t}{23}+111\cdot \arctan\left(37\cdot \frac{t}{294}\right)\right)+1}\right)}^{-1}$$

Table 2 offers an overview of the used model parameters aligned with the literature data [15,16,17,18,19,20].

Since the results depended heavily on the chosen LKB parameters, we also calculated the NTCPs for parameters varying by ±5%. The resulting NTCPs differed on average by less than 1%.

We created box and whisker plots to present the distribution of the NTCPs for the lungs, heart and esophagus. To appraise the NTCP differences between the treatment planning strategies, a Wilcoxon matched pairs test was applied. A *p* value below 0.05 was considered to indicate a statistically significant difference. The statistical analysis was performed with SPSS28.

## 3. Results

### 3.1. DVH Metrics

The volume of the contoured PTV ranged from 56 to 2600 cc (median: 496 cc), and the volumes of the heart and both lungs differed from 317 to 1152 cc (median: 691 cc) and 1151 to 5226 cc (median: 2433 cc), respectively. The mean Paddick conformity index for the APPA plans was 0.34, and 0.88 for the VMAT plans. Figure 1 shows an example of the dose distributions in two plan variants for a male patient in our cohort.

Table 3 shows the DVH metrics for heart, lungs and esophagus in the two planning variants, with the median value of each variable presented. Regarding the heart, the APPA plans showed lower Dmedian and dose levels at 1% of the organ volume (D1%), whereas VMAT planning resulted in lower Dmean and volume levels, which received 30 Gy or more (V30Gy). Regarding both lungs, the APPA planning resulted in lower Dmedian, but not the other shown metrics (Dmean, V20Gy and D1%). For the esophagus, VMAT planning resulted in significantly lower Dmean, Dmedian, and V1% values compared with APPA planning.

### 3.2. Estimated NTCP

Figure 2 and Table 4 show the results for APPA versus VMAT planning for the study cohort with different NTCPs. Mean and median as well as the maximum NTCPs for all organs were smaller for the VMAT planning (Table 4). No plan resulted in an NTCP greater than 5% for pericarditis. Risk for pneumonitis and impaired pericardial perfusions above 5% was only seen in APPA planning, but not in VMAT planning. Risk for increased cardiac mortality above 5% was seen in five APPA plans and four VMAT plans. Furthermore, most of the plans showed an increased NTCP for esophagitis for APPA planning versus VMAT planning. A significant difference in mean NTCPs when comparing APPA and VMAT planning was seen for increased cardiac mortality, pneumonitis and esophagitis.

### 3.3. Correlation Between DVH Metrics and NTCPs

For the NTCP concerning heart toxicities, no significant correlation between the volume of the heart and any NTCP in APPA or VMAT planning was seen. In contrast, the PTV size significantly correlated positively with the risk for pericarditis and increased pericardial perfusion in APPA planning, as well as increased cardiac mortality in both planning variations. The NTCP for pneumonitis correlated positively with the Dmean, Dmedian, V20Gym and D1% of the lungs in the APPA planning and with the PTV size as well as the Dmean, Dmedian and V20Gy for the lungs in the VMAT planning. The total volume of the lungs did not show significant correlation to the NTCPs in APPA or VMAT plans.

For esophagitis, the PTV volume, Dmean, Dmedian and V1% correlated positively with the NTCP values in both variants.

## 4. Discussion

The use of RT treatment in patients with PMBCL after responding to chemotherapy is a point of discussion as these patients are mostly young, and good oncological outcomes have been seen with intensive chemotherapy without RT or a watch-and-wait strategy with serial PET-CT imaging [8,21]. However, as we lack long-term data from these rare patients, a solid base for discussion is needed to weigh the risks and benefits of RT in this cohort. The presented patient cohort showed a broad range of PTV sizes (ranging from 56 to 2600 cc), reflecting the varying sizes of residual lymphoma bulks after systemic treatment. We compared two different plan variants for each patient to identify possible risk factors for increased NTCPs and optimal planning strategies for future patients.

The gathered data therefore serve as a valid base for evaluation of the metrics contributing to increased toxicities. We calculated the NTCPs for the lungs, heart and esophagus, as these OARs experience the highest doses during RT. As reported in the Methods section, we saw small divergences in the results when altering the NTCP model parameters. Nonetheless, one has to keep in mind that different NTCP model parameters can lead to different outcomes. As for our purpose of comparing plan variants and general risks of toxicities, we think the chosen parameters established before are adequate for use.

As previously described for patients with Hodgkin disease, cardiovascular toxicities can significantly contribute to the mortality of patients suffering from lymphoma with extension to the mediastinum [22,23]. Heart toxicities like pericardial disease or conduction disorders are induced by interstitial fibrosis or accumulation of myofibroblasts, which narrow arterial volumes [24]. Overall, multiple patient- and treatment-associated factors like smoking, diabetes, radiotherapy or anthracycline-containing chemotherapy have been identified as contributors to heart toxicity [23]. A recent dosimetric post hoc analysis of the British Rapid trial for Hodgkin’s lymphoma showed that the chemotherapy-associated increase in cardiovascular disease surpassed radiotherapy for most patients [25]. Regarding primary mediastinal B-cell lymphoma, one pivotal study failed to report the details on long-term cardiac toxicity [10].

For radiotherapy, a differential dose-side effect response has been described for various cardiac substructures [26]. The risk of valvular disease was increased after radiation doses above 30 Gy in 2 Gy fractions [25], and for coronary heart disease, a linear dose–response relationship was described [27]. The dose burden for the heart in mediastinal lymphoma RT may differ between patients, depending to their PTV sizes. However, the presence of a bulky tumor in itself is already a risk factor for an increased cardiac load [28], and the extend of a lymphoma mass toward critical heart structures might be as well [29]. As this was present in all patients in this study, special attention must be given to balancing the dose load on the heart.

In our investigation, the mean and median doses as well as the volume of the heart receiving 30 Gy or more (V30Gy) showed significant correlation with the NTCPs for pericarditis, increased cardiac mortality and impaired perfusion. This was the case for both APPA and VMAT plans and in line with the literature described for Hodgkin disease radiation cited above [28]. Only the D1% of the heart contour did not significantly correlate with the NTCPs for cardiac toxicities in either version. This might be due to the target dose of 40 Gy in this study, leading to D1% values of 40.4 and 40.4 Gy and D30% values of 22.0 and 11.3 Gy for the APPA and VMAT plans, respectively, which might not be associated with long-term toxicity [26,30]. None of the presented plans surpassed the commonly recommended constraints for the heart (e.g., V25Gy < 26% in the QUANTEC data or Dmean of 30 Gy in recommendations by the ILROG) [31,32].

Even though the heart’s Dmean and D1% were similar in APPA and VMAT planning, and the Dmedian was higher in the VMAT plans, the NTCP for increased cardiac mortality was significantly higher in the APPA vs. VMAT plans. This might hint at the importance of the V30Gy for this toxicity, which underlines the rather strong volume effect here [30,33]. In total, the median risk for impaired cardiac mortality was still was quite low in both planning types (0.03 versus 0.01%), not exceeding the observed value for cardiac toxicity in a large group of mediastinal lymphoma patients after receiving intensified chemo-immunotherapy (2.4%) [34].

Although the use of dose constraints for the lungs is widespread, conflicting data on the predictive value of DVH metrics for radiogenic pneumonitis exist [35]. The development of pneumonitis seems to be correlated with the mean lung dose as well as the maximum dose and the volume receiving more than 20 Gy (V20Gy) [18]. Nevertheless, clinical aspects like lung diseases and concurrent chemotherapy may also influence the development of side effects [35]. The association between clinically relevant pneumonitis and V20Gy aligns with findings from larger cohort studies [36] but might be of less impact than applied chemotherapy or the prevalence of COPD [37]. The model parameters used in our study were successfully compared to real-world outcomes of pneumonitis by Marks et al. [38]. We saw a significantly lower risk for pneumonitis in our VMAT plans compared with the APPA plans, even though the median NTCP was still low in both groups (0.03 versus 0.01%). This aligns well with other biophysical analyses of lymphoma RT planning and also real-world data from large cohorts [11,39]. Nevertheless, even if the risk for pneumonitis seems to be low for most patients, 20% of the APPA plans resulted in an estimated toxicity which exceeded a 5% complication probability. The advantage found for VMAT planning might be due to the high volume effect of the lungs, which was mirrored by the rather high n value we used in our model calculation [30].

In light of the fact that systemic therapy and RT were highly successful in the group of young lymphoma patients, attempts are being made to reduce both long-term and current toxicities by varying RT techniques an doses in order to maintain quality of life [40]. One such acute toxicity is esophagitis, which might be associated with pain, difficulty swallowing, impaired nutritional status and a decrease in quality of life (QoL) [41,42].

Roers et al. showed a significant impact of dose reduction on the esophageal NTCP in their planning study, showing 30 Gy compared with 40 Gy for thoracic lymphomas [39]. The dose reduction led to a decrease not only in the NTCP for esophagitis but also the NTCP for nausea, with both indicating a gain in QoL for such patients by reducing the esophageal dose burden. Furthermore, the authors demonstrated in a real-life cohort that the patients experienced nausea or dysphagia even at 0 Gy, suggesting that these symptoms may be influenced by prior treatment or the lymphoma itself. Reducing the dose for PMBCL was not tested in our planning study, yet varying the dose distribution through the use of VMAT planning led to significant decreases in DVH metrics and NTCP respectively (6.5 vs. 4.8%). Since RT for mediastinal lymphoma seems to lead to an inevitable dose burden on the esophagus, it is still remarkable that the use of VMAT versus APPA led to a significant reduction in the estimated NTCP. The proportion of plans exceeding the threshold of 5% risk for esophagitis was 50% lower for VMAT planning. The median dose burden achieved here with VMAT was favorable compared with the values reported in previous proton therapy studies [43].

## 5. Limitations

This study offers the first details on the impact of radiation planning on the estimated toxicities in PMBCL. The small cohort size may have led to distortions, but given the low incidence of this disease, the number of patients included still appears to be substantial. The use of a biophysical model like this did not consider preexisting morbidities, which can significantly impact heart and lung toxicity. The impact on induction from secondary malignancies could not be calculated with the used LKB model, which is important as the favored IMRT and VMAT techniques have been suspected to increase the risk of secondary malignancies due to the raised low-dose exposure (low-dose bath) in a healthy body [20]. In addition, some advanced techniques like proton therapy were not considered amongst the patients analyzed in this study. Proton therapy may enable breast sparing and can be especially beneficial for young female patients [32].

In the future, estimating and considering the corresponding risks of multimodal treatment regimens with exact descriptions of each modality will be of pivotal importance.

In this context, attention should also be paid to the interaction with immunotherapy, which has recently shown promising results in the treatment of PMBCL and other lymphomas but is likely to lead to an increased risk of pneumonitis in the interaction with radiation [44,45].

## 6. Conclusions

Our study showed decreased risk of different NTCPs in thoracic OARs for VMAT planning versus APPA planning for acute (esophagitis) and long-term (cardiac toxicities and pneumonitis) toxicities. The use of an IMRT technique like VMAT showed advantages for several DVH metrics in OARs and should therefore be recommended for radiation treatment of PMBCL. Future studies should investigate the effects described here in a prospective manner.

## Figures and Tables

**Figure 1 cancers-16-04265-f001:**
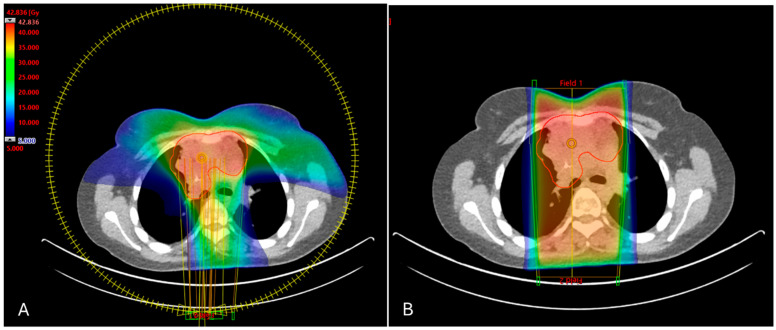
Example dose distributions for two plan variants for a male patient in our patient cohort. (**A**) VMAT plan. (**B**) APPA plan.

**Figure 2 cancers-16-04265-f002:**
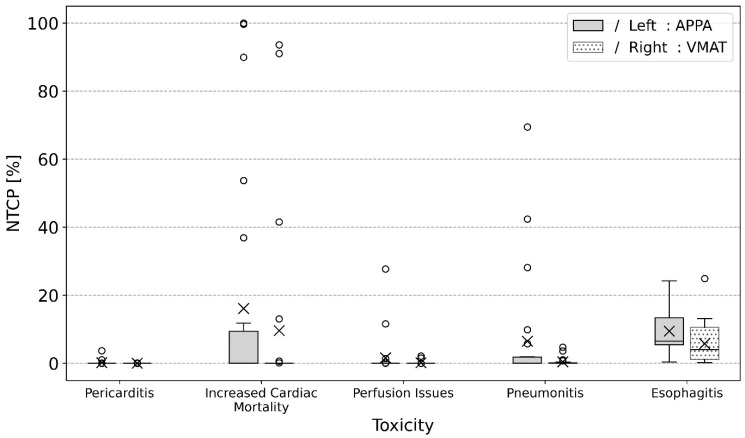
NTCP values for pericarditis, increased cardiac mortality, heart perfusion issues, pneumonitis and esophagitis for APPA vs. VMAT planning. Each boxplot displays the median (central line), mean (cross) and range between the 5th and 95th percentiles (whiskers), with any plotted outliers shown as individual points.

**Table 1 cancers-16-04265-t001:** Characteristics of patients with CT data included in this planning study.

Patients’ Characteristics		
**Sex**	male	n = 14
	female	n = 11
**Age**	mean: 34 years	range: 19–62 years
**Lymphoma stage at diagnosis**	1	n = 3
	2	n = 8
	3	n = 3
	4	n = 10
	extranodal manifestation	n = 17
	unknown	n = 1
**Chemotherapy regime**	daEPOCH-R	n = 15
	R-CHOEP	n = 8
	R-CHOEP-Brut	n = 2
**Further therapeutic scheme**	radiation	n = 6
	follow-up	n = 19

**Table 2 cancers-16-04265-t002:** NTCP model parameters used in this evaluation.

NTCP	Pericarditis	Increased Cardiac Mortality	Perfusion Issues	Pneumonitis	Esophagitis
n	0.5	0.5	0.5	0.87	0.44
TD50	48	29	41.9	24.5	51.5
m	0.1	0.1	0.1	0.18	0.32

**Table 3 cancers-16-04265-t003:** DVH metrics for all APPA and VMAT plans in this study. Values present the medians for the whole cohort of 25 patients, with interquartile ranges in brackets. * Statistical significance in the Wilcoxon test. Dmean = mean dose; Dmedian = median dose; VxGy = volume of respective organ which received x Gy; Dx% = dose which comprised x% of the respective organ.

		APPA	VMAT	*p*
**Heart**	**Dmean (Gy)**	11.0 (13.06)	9.8 (10.52)	0.19
	**Dmedian (Gy)**	2.2 (14.41)	2.6 (9.32)	<0.01 *
	**V30Gy**	22.0 (28.03)	11.3 (17.78)	<0.01 *
	**D1%**	40.3 (1.93)	40.4 (1.17)	0.3
**Lungs**	**Dmean (Gy)**	8.2 (10.04)	7.3 (7.09)	<0.01 *
	**Dmedian (Gy)**	2.16 (4.91)	3.8 (5.15)	0.037 *
	**V20Gy**	16.2 (23.46)	10.8 (15.56)	<0.01 *
	**D1%**	40.0 (1.91)	38.5 (5.04)	<0.01 *
**Esophagus**	**Dmean (Gy)**	18.6 (13.31)	14.8 (18.69)	<0.01 *
	**Dmedian (Gy)**	10.7 (35.64)	9.9 (31.0)	<0.01 *
	**D1%**	39.6 (1.72)	38.8 (10.52)	0.01 *

**Table 4 cancers-16-04265-t004:** Median NTCP values for the whole cohort (n = 25) for the organs heart, lungs and esophagus with APPA and VMAT planning, including the rate of cases in which the threshold value of 5% estimated toxicity was exceeded. * indicates significance.

Clinical Endpoint NTCP	Technique	Median %	Mean %	Minimum %	Maximum %	Proportion of Plans Resulting in an NTCP >5% (Total Number of Plans = 25)	*p* Value (Wilcoxon Test)
Pericarditis	APPA	0.0	0.19	0.0	4.0	0.0	0.29
	VMAT	0.0	0.01	0.0	0.0	0.0	
Increased cardiac mortality	APPA	0.03	16.2	0	100	0.28	<0.001 *
	VMAT	0.01	9.6	0	0	0.16	
Impaired cardiac perfusion	APPA	0.0	1.64	0	28	0.08	0.23
	VMAT	0.0	0.15	0	2	0.0	
Pneumonitis	APPA	0.03	6.42	0	69	0.2	<0.001 *
	VMAT	0.01	0.46	0	5	0.0	
Esophagitis	APPA	6.5	9.4	0.4	24.3	0.8	<0.001 *
	VMAT	4.8	5.8	0.0	25.0	0.4	

## Data Availability

The data presented in this study are available in this article, and raw data are available upon request.

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
