# Peer review of "Minimizing Long-Term Toxicities for Patients with Primary Mediastinal B-Cell Lymphoma Undergoing Modern Radiotherapy: Results from a Monocentric Biophysical Risk Evaluation"

_cancers, 2024, doi:10.3390/cancers16244265_

Round 1

Reviewer 1 Report

Comments and Suggestions for Authors I would like to thank the Editors for the possibility to review this manuscript. The topic examined in this paper is of interest and adds to current literature regarding the risk of radiation induced toxicities in patients treated   for mediastinal lymphomas. The paper is overall well written and methodologically sound, nonetheless, I would suggest to implement the following revisions.   Abstract: it is stated that mean heart dose is lower for APPA, and this is in contrast with the actual results   Introduction "Some authors even propagate advantages" please rephrase for clarity.   Methods: the guidelines and process adopted to delineate the CTV should be specified. Moreover, CTV-PTV margins are not clear: di they differ from patient to patients? Why using margins as large as 2 cm? Could you please specify the choice to use median dose and its clinical expected impact?   Discussion The impact of disease volume is correctly explicitated. I would also specify the impact of disease localization on some OARs (e.g. disease extending to the lower mediastinum for heart DVHs).  Immunotherapy has recently demonstrated its effectiveness for the treatment of PBCL (https://doi.org/10.1182/bloodadvances.2023010254). Combination with thoracic radiotherapy conceptually could increase the risk of pneumonitis, nonetheless preliminary data regarding combined treatment for  lymphoma (https://doi.org/10.3324/haematol.2023.284689 doi: 10.1016/j.clml.2021.09.005 doi: 10.1200/JCO.22.02355) are reassuring. I would add a paragraph on this topic. In the limitations, I would expand and stress the importance of treatment induced second malignancies. I would briefly summarize the potential impact of IMRT compared with less conformal techniques, on the basis of recent reviews and studies. I would also stress the higher clinical acute and subacute toxicity burden of chemotherapy intensification, and the fact that chemotherapy bears as well an even potentially larger risk of cardiotoxicity and SMNs    

Author Response

Thank you for reading and improving our paper. We tried to improve everything according to your recommendations.

Yes

Can be improved

Must be improved

Not applicable

Does the introduction provide sufficient background and include all relevant references?

( )

(x)

( )

( )

Is the research design appropriate?

(x)

( )

( )

( )

Are the methods adequately described?

( )

(x)

( )

( )

Are the results clearly presented?

( )

(x)

( )

( )

Are the conclusions supported by the results?

( )

(x)

( )

( )

Comments and Suggestions for Authors

I would like to thank the Editors for the possibility to review this manuscript. The topic examined in this paper is of interest and adds to current literature regarding the risk of radiation induced toxicities in patients treated   for mediastinal lymphomas. The paper is overall well written and methodologically sound, nonetheless, I would suggest to implement the following revisions.

Abstract: it is stated that mean heart dose is lower for APPA, and this is in contrast with the actual results

  • Thank you for that remark. We corrected this sentence.

Introduction "Some authors even propagate advantages" please rephrase for clarity.

  • We rephrased this sentence.

Methods: the guidelines and process adopted to delineate the CTV should be specified. Moreover, CTV-PTV margins are not clear: di they differ from patient to patients? Why using margins as large as 2 cm?

  • The contouring was aligned to the guideline on contouring of DLBCL by Oertel et al. (See references). We adapted our description here to improve clarity.

Could you please specify the choice to use median dose and its clinical expected impact?

  • Thank you for this question. We see that for the heart, the median dose differs strongly from the mean dose. In clinical practice, we need to consider that high mean doses might be an expression of high doses at small volumes whereas the majority of the organ remains with lower doses. We think the presentation of the median underlines the value of regarding several DVH parameters when comparing plans in general as it represents a more comprehensive picture of the dose distribution.

Discussion

The impact of disease volume is correctly explicitated. I would also specify the impact of disease localization on some OARs (e.g. disease extending to the lower mediastinum for heart DVHs). 

  • We added a sentence and respective references here.

Immunotherapy has recently demonstrated its effectiveness for the treatment of PBCL (https://doi.org/10.1182/bloodadvances.2023010254). Combination with thoracic radiotherapy conceptually could increase the risk of pneumonitis, nonetheless preliminary data regarding combined treatment for  lymphoma (https://doi.org/10.3324/haematol.2023.284689 doi: 10.1016/j.clml.2021.09.005 doi: 10.1200/JCO.22.02355) are reassuring. I would add a paragraph on this topic.

  • Thank you very much. We included the respective sentences.

 In the limitations,

I would expand and stress the importance of treatment induced second malignancies. I would briefly summarize the potential impact of IMRT compared with less conformal techniques, on the basis of recent reviews and studies.

  • We added a passage about that topic to our limitation section.

I would also stress the higher clinical acute and subacute toxicity burden of chemotherapy intensification, and the fact that chemotherapy bears as well an even potentially larger risk of cardiotoxicity and SMNs    

  • Thank you for this comment. We added a paragraph concerning this topic in the discussion.

Reviewer 2 Report

Comments and Suggestions for Authors

This is a planning study to compare NTCP values for AP-PA to VMAT for primary mediastinal B-cell lymphoma to determine if one technique should be preferred to minimize side effects from radiation. One major weakness is that this is only a planning study and no patient outcomes are given, so it is purely theoretical. Also, the manuscript is weak on many details to help understand the study. Specific comments are numbered below.

1. Abstract line 20: in a couple of places it says that this study was prospective. Planning studies are usually retrospective, as that allows using a larger patient cohort. Why was this study done prospectively? And if it was prospective, why is there no information on which plan was used to treat the patients and what the toxicity outcomes were for treatment? The Methods do not mention any IRB approval for human experimentation or consent, which should always be required for a prospective trial. This is a serious matter that needs to be clarified.

2. Methods line 70: Information on the criteria for accepting patients into this prospective trial needs to be included. If it is actually retrospective, not prospective, the authors still need to provide information on how they selected patients for this study, such as how many years it took to acquire this number of patients. Were there any limits on the age of a patient to be included in the study? Also, some information on the patient radiotherapy setup should be given, such as what immobilization devices were used, were they treated with arms up or arms down, and what VMAT arc arrangement was used for planning (based on Figure 1, it may have been a full arc with avoidance sectors to spare the lungs?)

3. Methods line 100: the "n" parameter is related to the volume effect, which is not well-described by calling it a tissue sensitivity parameter. See for example https://pmc.ncbi.nlm.nih.gov/articles/PMC10035357/ and references therein. Using only one set of NTCP parameters, given in Table 1, when others exist in the published literature is also a huge limitation of this study, as the results could change with different parameters. This should be added to the Discussion.

4. Results line 125: the first table in the results should be a list of patient characteristics, such as Table 2 in https://pmc.ncbi.nlm.nih.gov/articles/PMC10035357/. The volumes of PTV, heart, lungs, and esophagus should be given in that table, along with clinical characteristics of the patients such as age, gender, disease stage, prior treatment summary, etc. The volumes of the heart and lungs vary much more widely than I would expect to see for adult patients over the age of 18, so I assume some of these patients were pediatric. These details should be provided so others reading the paper know what patients were included in the study.

5. Results line 127: Paddick conformity index has not been defined or referenced.

6. Figure 1: in this image, it appears the patient is on a curved couch top. This would not be allowed for treatment on a TrueBeam linac. Is this the image of the patient used for treatment? Or is this a diagnostic image used for the planning study? If this was a prospective study, which plan was used for treatment for each patient and how was that determination made?

7. Table 2: these values are averaged over all 25 patients, correct? This should be explicitly stated and a standard deviation given for each value averaged over the cohort. Some of the results do not appear to be statistically significantly different, such as the Heart D1% which is 40.3 Gy for AP-PA and 40.4 Gy for VMAT, so these p-values should all be checked again for errors. 

8. Figure 2: what do the "x" markers indicate in the figure? The caption says the mean is displayed as a square but I do not see squares in the figure.

9. Table 3: the "Maximum" results for VMAT pericarditis and increased cardiac mortality appear to be incorrect, as they are shown to be 0 but the mean values are non-zero. Please check all of these values again for accuracy. Because of the way the results were analyzed, by comparing the proportion of plans where the calculated NTCP values exceeded a threshold of 5%, the results may be very susceptible to uncertainties in the exact LKB parameters used. If the results were instead compared by the change in NTCP values between the two plans, it may be less impacted by the exact LKB parameters used.

10. Limitations line 252: as mentioned previously, the uncertainty in the LKB parameters could have a large impact on the results and is not controlled for in any way in the current study.

Author Response

Thank you for your valuable comments. We tried to answer every single question and to adapt our text and data accordingly.

Does the introduction provide sufficient background and include all relevant references?

(x)

( )

( )

( )

Is the research design appropriate?

( )

(x)

( )

( )

Are the methods adequately described?

( )

( )

(x)

( )

Are the results clearly presented?

( )

( )

(x)

( )

Are the conclusions supported by the results?

( )

(x)

( )

( )

Comments and Suggestions for Authors

This is a planning study to compare NTCP values for AP-PA to VMAT for primary mediastinal B-cell lymphoma to determine if one technique should be preferred to minimize side effects from radiation. One major weakness is that this is only a planning study and no patient outcomes are given, so it is purely theoretical. Also, the manuscript is weak on many details to help understand the study. Specific comments are numbered below.

Abstract

  1. Abstract line 20: in a couple of places it says that this study was prospective. Planning studies are usually retrospective, as that allows using a larger patient cohort. Why was this study done prospectively? And if it was prospective, why is there no information on which plan was used to treat the patients and what the toxicity outcomes were for treatment? The Methods do not mention any IRB approval for human experimentation or consent, which should always be required for a prospective trial. This is a serious matter that needs to be clarified.

--> Thank you for this comment. We clarified our study design concerning that aspect by adapting our methods section: Indeed, the planning study evaluated plan variants for a cohort of PMBCL patients treated at our institution with chemotherapy, and some of them with additional radiotherapy. All radiation plans were designed exclusively for this planning study, what led us to the term “prospective”. As addition of radiotherapy is a point of discussion in every case of PMBCL between radiation oncologists and oncologists, we wanted to provide insight in possible toxicity rates for a cohort of patients with different anatomic structures as displayed in their PET CT Scans after chemotherapy. Our study can help during therapy decisions for future patient with this rare disease.

--> We deleted the word prospective as it is confusing and as you stated not correct.

Methods

  1. Methods line 70: Information on the criteria for accepting patients into this prospective trial needs to be included. If it is actually retrospective, not prospective, the authors still need to provide information on how they selected patients for this study, such as how many years it took to acquire this number of patients. Were there any limits on the age of a patient to be included in the study? Also, some information on the patient radiotherapy setup should be given, such as what immobilization devices were used, were they treated with arms up or arms down, and what VMAT arc arrangement was used for planning (based on Figure 1, it may have been a full arc with avoidance sectors to spare the lungs?)

--> Thank you for this comment. As described before, the term prospective was deleted here to avoid confusion. We added a table of patient characteristics in the methods section, since the patient characteristics were not a result of the study but the basic conditions of the investigation.

--> 6 of 25 patients actually received radiation treatmen. As the planning study starts at the PET CT scan after completion on chemotherapy and none of the patient was actually treated with one of the hereby designed plans, details on the radiation setup are not mentioned.

--> We added a sentence on the VMAT arrangement.

  1. Methods line 100: the "n" parameter is related to the volume effect, which is not well-described by calling it a tissue sensitivity parameter. See for example https://pmc.ncbi.nlm.nih.gov/articles/PMC10035357/ and references therein. Using only one set of NTCP parameters, given in Table 1, when others exist in the published literature is also a huge limitation of this study, as the results could change with different parameters. This should be added to the Discussion.

--> Thank you for that remark. We changed the phrase in the methods section.

--> To estimate uncertainties in the chosen LKB parameters, all NTCP values were computed while varying n,m, and TD50 by +-5% - with variations in TD50 having the largest effect (1% difference on average). Overall, out analysis showed that the uncertainty of the LKB parameters does not change the trend oft he results in a significant way. The chosen LKB parameters come from peer-review publications. Similarly to our uncertainty analysis, absolute values will vary with different LKB parameters – however not in a meaningful enough scale as to change the conclusions drawn from the calculations presented in the paper.

--> We added a phrase in the discussion section.

Results

  1. Results line 125: the first table in the results should be a list of patient characteristics, such as Table 2 in 4. https://pmc.ncbi.nlm.nih.gov/articles/PMC10035357/. The volumes of PTV, heart, lungs, and esophagus should be given in that table, along with clinical characteristics of the patients such as age, gender, disease stage, prior treatment summary, etc. The volumes of the heart and lungs vary much more widely than I would expect to see for adult patients over the age of 18, so I assume some of these patients were pediatric. These details should be provided so others reading the paper know what patients were included in the study.

--> We added a table with patients` characteristics to create an overview of the cohort.

  1. Results line 127: Paddick conformity index has not been defined or referenced.

--> We added a phrase in the methods section.

  1. Figure 1: in this image, it appears the patient is on a curved couch top. This would not be allowed for treatment on a TrueBeam linac. Is this the image of the patient used for treatment? Or is this a diagnostic image used for the planning study? If this was a prospective study, which plan was used for treatment for each patient and how was that determination made?

--> As described before, our planning was performed on the diagnostic imaging.

  1. Table 2: these values are averaged over all 25 patients, correct? This should be explicitly stated and a standard deviation given for each value averaged over the cohort. Some of the results do not appear to be statistically significantly different, such as the Heart D1% which is 40.3 Gy for AP-PA and 40.4 Gy for VMAT, so these p-values should all be checked again for errors. 

--> Thank you for this input. We re-checked our results and found no irregularities here. Indeed, the heart D1% is not significantly different between the plan variant, as the p-value of 0.3 indicates.

--> We stated that all values are median values over the whole cohort in our text and added this information additionally in the caption.

--> Median values are commonly not supplemented with standard deviations. We added interquartile ranges instead.

  1. Figure 2: what do the "x" markers indicate in the figure? The caption says the mean is displayed as a square but I do not see squares in the figure.

--> Thank you for this hint. We adapted our caption here.

  1. Table 3: the "Maximum" results for VMAT pericarditis and increased cardiac mortality appear to be incorrect, as they are shown to be 0 but the mean values are non-zero. Please check all of these values again for accuracy. Because of the way the results were analyzed, by comparing the proportion of plans where the calculated NTCP values exceeded a threshold of 5%, the results may be very susceptible to uncertainties in the exact LKB parameters used. If the results were instead compared by the change in NTCP values between the two plans, it may be less impacted by the exact LKB parameters used.

--> Thank you for this remark. We noticed that by omitting further decimal places, the value was not clearly displayed here.

--> Indeed, the choice of the 5 %-threshold bears disadvantages but for the pragmatic clinical use it is easy to understand differences/advantages between plan variants as clinical radiation oncologists know the 5 %-threshold as a limit of toxicity.

  1. Limitations line 252: as mentioned previously, the uncertainty in the LKB parameters could have a large impact on the results and is not controlled for in any way in the current study.

--> Compare to comment 3.
